# Effect of Interlayer Bonding Temperature on the Bending Properties of Asphalt Concrete Core Wall

**DOI:** 10.3390/ma16114133

**Published:** 2023-06-01

**Authors:** Qunzhu Han, Haoyu Dong, Yingbo Zhang, Taotao Gao, Ge Song, Shanwang Wang

**Affiliations:** 1State Key Laboratory of Eco-Hydraulics in Northwest Arid Region, Xi’an University of Technology, Xi’an 710048, China; hanqz@xaut.edu.cn (Q.H.); zhangyb68@163.com (Y.Z.); 18710703025@163.com (T.G.); gsong1369@163.com (G.S.); 18238606873@163.com (S.W.); 2School of Civil Engineering and Architecture, Xi’an University of Technology, Xi’an 710048, China

**Keywords:** asphalt concrete core wall, combined surface, temperature, bending properties, construction quality

## Abstract

In the construction process of an asphalt concrete impermeable core wall, the interlayer bonding of the core wall is the weak link of the core wall structure and also the focus of construction, so it is important to carry out research on the influence of interlayer bonding temperature on the bending performance of an asphalt concrete core wall. In this paper, we study whether asphalt concrete core walls could be treated with cold-bonding by fabricating small beam bending specimens with different interlayer bond temperatures and conducting bending tests on them at 2 °C. The effect of temperature variation on the bending performance of the bond surface under the asphalt concrete core wall is studied through experimental data analysis. The test results show that the maximum value of porosity of bituminous concrete specimens is 2.10% at lower bond surface temperature of −25 °C, which does not meet the specification requirement of less than 2%. The bending stress, strain, and deflection of bituminous concrete core wall increase with the increase of bond surface temperature, especially when the bond surface temperature is less than −10 °C. If the lower bonding surface temperature is less than 10 °C, the upper layer of asphalt mixture with large grain size aggregate cannot be effectively buried in the low bond surface, resulting in flat fracture and brittle damage to the specimen, which is detrimental to construction quality; therefore, the bonding surface should be heated so that the temperature of the bottom bonding surface is 30 °C. If the lower bonding surface temperature is 10 °C or above, no heating is required.

## 1. Introduction

Hydraulic asphalt concrete core walls, panels, and geomembranes are the main impermeable structures for pumped storage power plants [1,2,3], among which asphalt concrete core wall dams are widely used in water conservancy projects because of their simple structure, fast construction, and excellent impermeability with good deformation and seismic resistance [4,5,6]. During the construction process of an asphalt concrete impermeable core wall, the bond between core layers is the weak link of the core wall structure [7,8] and thus also the focus of construction, because hydraulic asphalt concrete has a strong temperature sensitivity [9,10,11]. Especially in low-temperature environments [12,13,14], asphalt concrete is most affected by temperature. Its construction is usually carried out at room temperature, but in some low-temperature environments, if the base temperature of asphalt concrete is too low, the construction difficulty will increase and the quality of construction will be difficult to guarantee. Therefore, China stipulates that the temperature of the combined surface of the asphalt concrete core wall cannot be lower than 70 °C [15].

The surface temperature of the hydraulic asphalt concrete core wall bond is mostly inconsistent within each country, and some of the German asphalt roll concrete dams, when paved, require the subgrade surface temperature to be in excess of 25 °C; if this is not the case, heating of the subgrade surface is required [16]. In 1989, for example, for the construction of the German Schmalwasser (Schmalwasser) asphaltic concrete heart wall dam, depending on the construction requirements, the surface temperature of the bottom layer needed to be greater than 25 °C before the top layer of the mixture was paved; if this was not the case, the bonding surface of the asphalt concrete core wall had to be heated [17]. The Yeller asphalt concrete core wall dam was completed in China in 2005 and required an interlayer heating temperature of no less than 70 °C, but it had been successfully built in a low-temperature environment, and observations during and after construction showed that the asphalt concrete core wall worked well [18]. In recent years, a project in Xinjiang, China [19] and a project on the construction of an asphalt concrete core wall in Shiazi Mountain, Yunnan [20] used the top layer of hot material to raise the temperature of the bonding layer, but more projective construction used the intermediate layer heating specification to treat the bonding surface [21].

Researchers are currently studying the interlayer bonding performance of asphalt concrete core walls at a variety of temperatures. Cai Qian [22], for example, concluded that the flexural strength and flexural deformation of the asphalt concrete bond did not vary greatly and satisfied the quality requirements by not heating the asphalt concrete bond during the construction of the simulated core wall. Wan Lianbin [23] showed through field tests that, when the temperature of the bond surface of asphalt concrete core wall is reduced to 40 °C, the bond surface does not need to be heated, and the bond surface quality can satisfy the conditions. On the basis of an indoor simulation test, Zhu Xichao [24] concluded that the top and bottom layers of asphalt concrete can be compactly combined when the bond surface temperature of the asphalt concrete core wall is 30 °C, and there is no significant decrease in the mechanical properties of asphalt concrete at the bonding surface. Through field construction, Gao Taotao [25] found that the bond surface bending performance of the asphaltic concrete core wall did not have a strong effect as the bond surface temperature fell to 0 °C. For a core wall made of asphalt concrete with a lower bond area temperature of −20 °C or lower, Qiao Yong and Du Lihong [26] verified the construction of the core wall without heating the underlying asphalt concrete by carrying out an experimental investigation. Zhu Xichao [27] found that the porosity of the specimens was not able to meet the requirement of less than 2% in the specification when the surface temperature of the bond was brought down to −25 °C. Liu Rubo [28] and Fan Zhenjun [29] also tested the core wall made of asphalt concrete under a low-temperature environment and found that there was no obvious delamination at the surface of the bond, and that the position of the binding surface could not be distinguished by eye and binding was satisfactory.

Temperature regulation is a major challenge in the construction of hydraulic asphalt concrete fields. The majority of the aforementioned researchers’ studies on the bonding surface of an asphalt concrete core wall are performed at room temperature, although it is unclear whether cold, low-temperature binding meets the quality criteria.

Small beam bending specimens were prepared in this study by designing different bond-surface temperatures, by controlling the temperature of the binding surface (−25 °C to 70 °C), and by performing bending tests at 2 °C. The primary problem addressed in this study is verifying that the binding surface can be treated with cold binding, which may provide a benchmark for later construction and maintenance of asphalt concrete core walls in colder regions. It is therefore important to perform investigations into the effect of interlayer bond temperature on the flexural performance of asphalt concrete core walls.

## 2. Test Program

### 2.1. Test Content

The trabecular bending specimens were prepared by designing different joint surface temperatures and the bending test was carried out at 2 °C. The flexural strength of asphalt concrete and the strain corresponding to flexural strength and deflection across the middle span are obtained. Finally, stress strain analysis, fracture section analysis, and data comparison analysis are carried out according to the experimental results.

### 2.2. Testing Raw Materials

The asphalt model used in this test is Karamay 70 asphalt. The coarse aggregates were limestone aggregates, divided into three categories, according to 9.5–16, 4.75–9.5, and 2.36–4.75 mm; the fine aggregate was limestone artificial sand, 0.075–2.36 mm, and the filling was limestone mineral powder, up to 0.075 mm, ground by a laboratory ball mill [30].

### 2.3. Raw Material Testing

The raw material testing was based on SL 501-2010 “Design code of asphalt concrete facings and cores for embankment dams” [31]. The performance of the asphalt concrete impermeable layer is directly affected by the properties of raw materials, so the key to ensuring the performance of asphalt concrete is to choose the raw materials that meet the requirements. The main raw materials of hydraulic asphalt concrete are asphalt, coarse aggregate, fine aggregate, fillings, etc. The selection criteria and requirements of raw materials are introduced below, as well as the basic performance test analysis; the basic requirements are shown in Table 1, Table 2, Table 3 and Table 4.

(1)Coarse aggregates

Crude aggregate has a particle size of 19~2.36 mm and acts mainly as a skeleton in asphalt concrete, the quality of which is an important index of asphalt concrete strength. In this paper, limestone was used as coarse aggregate, which was divided into 5 particle size classes by broken sieve. The surface density, water absorption, adhesion to asphalt, crushing rate, and durability of the two coarse aggregates were tested and compared. The results of mass tests of the coarse aggregate are shown in Table 1.

(2)Fine aggregates

Fine aggregate particle size falls within 2.36–0.075 mm, and its main role is to lubricate the asphalt concrete, in order to provide fluidity for the asphalt concrete, so that the asphalt concrete at a certain temperature can better form the flow dynamic, be easy to pave, and have good compaction. The fine aggregate also fills the larger gaps between the coarse aggregate, making the aggregate more dense. In this paper, fine aggregate was prepared from hard limestone, and its surface density, water absorption, water stability grade, durability, and mud content were tested and compared. The results of quality inspection of fine aggregate are shown in Table 2.

(3)Filler

The role of filler is to fill the fine voids between coarse and fine aggregates to make the asphalt concrete aggregate more dense, which helps to better improve the impermeability properties of asphalt concrete. The filler comprises ground limestone as the raw material, and the main test indexes are surface density, water content, hydrophilic coefficient, and so on. The quality test results of the fillers are shown in Table 3.

(4)Asphalt

Asphalt is the core material of the asphalt concrete impermeable layer, and the quality of asphalt is related to its impermeable performance. At present, the comprehensive performance of asphalt concrete is mainly judged by three indexes: needle penetration, softening point, and ductility. Therefore, in this paper, the three major indicators of the selected asphalt (before film oven and after film oven) were tested. The asphalt quality test results are shown in Table 4.

The test results in Table 1, Table 2, Table 3 and Table 4 show that all the test indicators of the coarse and fine aggregates, fillers, and asphalt meet the requirements of the current specifications [31,32].

### 2.4. Material Matching Ratio

The design grading of mineral for hydraulic asphalt concrete is calculated using the improved Ding Purong formula based on Fuller’s (Fuller) formula.
(1)$$P_{i}=P_{0.075}+\left(100-P_{0.075}\right)\frac{{\left(d_{i}\right)}^{r}-{\left(0.075\right)}^{r}}{{\left(D_{max}\right)}^{r}-{\left(0.075\right)}^{r}}$$

In the Formula (1): *P_i_* is the passing rate of aggregate at *d_i_*, %; *d_i_* is the particle size of a grade, mm; *P*_0.075_ is the passing rate of aggregate when the sieve is 0.075 mm, %; *D_max_* is the maximum particle size of aggregate, mm; and *r* is the grading index.

The mineral grading index is 0.38, the oil to stone ratio is 7.2%, and the filler amount is 12%. The aggregate grading curve is shown in Figure 1.

The mineral aggregate gradation index refers to the proportion of mineral aggregate that constitutes each particle size of asphalt mixture. The oil-to-stone ratio means the weight percentage of the amount of asphalt and minerals in the asphalt mixture. The amount of filler refers to the amount of mineral powder used in the asphalt mixture to fill the voids of asphalt concrete in the overall mineral content.

## 3. Test Overview

### 3.1. Test Equipment

The test equipment mainly include the following: bending test machine (1 KN, 5 KN); motor controller (500 W); data collector (N); computer; pressure sensor (specification 50 N, 800 N); thermometer (−30 °C~70 °C ); cold box (−35 °C~24 °C ); glass water; and flame spray lamp.

The bending test uses the motor controller to control the motor at the bottom of the bending test instrument, drives the thermostatic waters support in the specimen to move upwards, and fixes the pressurized rod position, so that the support forms a three-point bending test on the asphalt concrete beam. The force applied to the middle of the specimen is transmitted from the rod to the upper pressure sensor and then transmitted to the data collector by the pressure sensor; the bending deformation in the middle of the specimen is transmitted to the data collector by the displacement sensor and finally stored in the computer in the form of electrical signals in the computer’s asphalt concrete bending test system.

### 3.2. Specimen Forming Method

The sample size for the trabecular bending test is a rectangular specimen of 250 mm × 35 mm × 40 mm as required by specification [32], which is formed mainly by compaction and cutting. The operation is as follows:(1)Aggregates of a certain ratio are heated in a high-temperature oven for 3 to 4 h at 170 °C;(2)The aggregate and bitumen in proportion to the bituminous concrete mixer are heated, stirring at 160 °C for 90 s to form an asphalt mixture;(3)In a mold used for the production of specimens sized 300 × 150 mm, samples are divided into four layers of 75 mm each in order to simulate the mass of interlayer adhesion under different test conditions. The lower layer of asphalt concrete is prepared, and compaction is performed 7 times each round, with a total of 15 rounds;(4)The temperature change rate is controlled to about 28.6% of the lower temperature limit of 70 °C specified in the construction specification [15] for low-level blends. The same mixture ratio of −25 °C, −10 °C, 10 °C, 30 °C, 50 °C, 70 °C is then laid on top of the mixture of 160 °C asphalt, with the same test preparation method as before;(5)The mold is removed after the test block has cooled (Figure 2);(6)The specimen is then cut into a small, curved specimen size of 250 mm × 35 mm × 40 mm, deviation ±1 mm, ±1 mm, ±2 mm, respectively, with a special cutter. The combined surface located at 125 mm along the length of the cut specimen is well marked.

### 3.3. Test Procedure

(1)The first step requires the sample to be treated at constant temperature, adjusting the thermostat according to the temperature required for the test. When the thermostat reaches the minimum underwater temperature of [31] 2 °C in China, the sample is put and activated at a set time, and after 4 h the sample temperature can meet the test temperature requirements with an error of ±0.5 °C.(2)The next step is to adjust the test machine. Place and adjust the test machine test piece support, with support distance of 200 mm, and ensure that the upper force rod lower indenter and the support are parallel and centered. Then, turn the motor until the indenter and support are in a horizontal direction and form a certain distance, in order to facilitate the placement of test pieces;(3)Next is to adjust the test specimen thermostat, the thermostatic water tank on the curved test stand, and the temperature of the glass water (specification: −40 °C) as required by the test. Measure the temperature of the water by thermometer and adjust the temperature in real time so that the temperature of the water is within ±1 °C of the test requirement;(4)The constant temperature and temperature-adjusted specimens are placed in the test machine at a height of 35 mm and width of 40 mm, provided that the temperature of the bath meets the test requirements (Figure 3);(5)The motor controller is adjusted so that the motor rotates at a certain speed with a strain rate of 1%/min, i.e., a span deformation rate of 1.67 mm/min. The load is applied between the specimens and data are automatically collected by computer.

### 3.4. Specimen Forming

Figure 4 presents the following: (1)When the lower layer of asphalt is applied with the bonding layer temperature of −25 °C and −10 °C, a large band gap appears at the binding surface and can obviously be seen to delaminate, denoting that the binding quality is poor.(2)When the lower asphalt mixture layer is applied with the 10 °C and 30 °C surface temperature setting, small voids appear on the surface and there is no clear delamination, thus denoting good quality.(3)When the lower asphalt mixture layer is applied with the surface temperature setting at both 50 °C and 70 °C, there is no visible delamination and there are no voids in the combined surface, thus denoting good quality.

## 4. Test Results and Analysis

According to DL/T 5362-2018, the tensile strength of asphalt concrete is calculated by Equation (2), and the maximum bending tensile strain is calculated by Equation (3). The stress–strain curves of asphalt concrete at different temperatures are produced separately from the experimental data.
(2)$$R_{b}=\frac{3LP_{b}}{2bh^{2}}$$
(3)$$\varepsilon_{b}=\frac{6hf}{L^{2}}\times 100\%$$

In the formula: *R_b_*—flexural strength of asphalt concrete, Mpa; *b*—width of the span-interrupted specimen, mm; *h*—height of the span-interrupted specimen, mm; *L*—span diameter of the specimen, mm; *P_b_*—specimen damage at the maximum load, N; *ε_b_*—maximum bending tensile strain of asphalt concrete, %; and *f*—the deflection of the span midpoint when the specimen is damaged, mm.

After the density and porosity of the sample are measured, three specimens are prepared according to the above test procedure at different bonding surface temperatures and the mean of the test results is used as the test result. When the difference between the maximum or minimum of three specimens and the mean of the test value exceeds 15%, the mean value shall be the average. When the maximum and minimum values of the three test samples differ by more than 15% from the median values, the test should be repeated. The relationship between the heating temperatures and the bending strength and tensile strength of the bonding surface at different heating temperatures is compared and analyzed. The experimental results are shown in Table 5 and the stress–strain curve is shown in Figure 5.

Density: The density of a material is the mass per unit volume of the material in an absolutely dense state.

Porosity: It is the percentage of pore volume in the block material to the total volume of the material in its natural state.
(4)$$P=\frac{V_{0}}{V}\times 100\%=\left(1-\frac{\rho_{0}}{\rho }\right)\times 100\%$$

In the formula: *P*—material porosity, %; *V*_0_—material volume in the natural state, or apparent volume, cm^3^; *ρ*_0_—material bulk density, g/cm^3^; *V*—absolute dense volume of material, cm^3^; and *ρ*—material density, cm^3^.

As can be seen from Table 5 and Figure 5, the density of trabecular bending specimens increases with the increase of the bond surface temperature and the porosity decreases with the decrease of the bond surface temperature. The maximum flexural strength, the maximum bending tensile strain, and the deflection in the span corresponding to the maximum flexural strength all increase with the increase of the bond surface temperature.

The maximum porosity of bituminous concrete specimens is 2.10% when the temperature of the lower bond surface is −25 °C, which does not meet the requirement of <2% in the design specification for hydraulic bituminous concrete. This is because −25 °C is below the asphalt brittle point, the upper layer of hot material poured into the temperature dissipates faster, and large particles of aggregate in the compaction molding are difficult to embed in the lower layer of material, so the density of the test piece in the compaction molding is difficult to achieve.

When the initial loading temperature of the lower bond surface is −25 °C and −10 °C, the force of the specimen is unstable, and force interruption and loading deflections are likely to occur. This phenomenon is due to poor adhesion between the upper and lower layers of the specimen and gaps in the bond surface, as shown in Figure 4. The bending stress–strain curve reflects poor bending performance, and there is a large unstable gap between the data peaks. This is mainly due to the poor adhesion between the upper and lower layers of the thermal materials and the brittleness damage after stress.

As can be seen from Figure 6, when the temperature of the lower bonding surface is −25 °C and −10 °C, the trabecular curved sample has a smoother cross section, indicating poor binding quality of the bonded surface, as there is no aggregate embedded in the larger particles in the lower asphalt concrete when the sample is intercalated. When the surface temperature of the lower bond is 10 °C, 30 °C, 50 °C, and 70 °C, the surface bond is rougher and the bond is good. This is because the upper 160 °C mixture will be present in the "exhaust" process during paving [33], so the contact portion of the upper asphalt mixture and the lower asphalt mixture will increase their lower temperature to 70 °C in 30 min, bringing the lower mixture down. It can be seen that different heating temperatures have a great influence on the interlayer binding quality below −10 °C.

In order to better analyze the effect of interlayer heating temperature on the bending performance, by using 70 °C as the study ontology, the stress ratio, strain ratio, and deflection ratio variation curves were constructed, and the effect of the interlayer heating temperature on the bending performance of asphalt concrete is shown in Figure 7.

Stress ratio is the ratio of stress measured at different temperatures to stress measured on bonding surface; strain ratio is the ratio of strain measured at different temperatures to strain measured on the bonding surface; and deflection ratio is the ratio of deflection measured at different temperatures to the deflection measured on the bonding surface.

As can be seen from Figure 7, the bending stress ratio, bending strain ratio, and deflection ratio are close to 1 at different temperatures, and the bond layer temperature increases at 2 °C. Overall, the ratio of all three is close to 1, with a smaller slope at temperatures of 10, 30, 50, and 70 °C in the bond layer, indicating good bond quality between layers.

## 5. Conclusions

In this paper, bending tests were conducted on asphalt concrete specimens in a 2 °C temperature environment, and the bending properties of −25 °C, −10 °C, 10 °C, 30 °C, 50 °C, and 70 °C interlayer bonding temperatures were analyzed. The results are concluded as follows:(1)There is a lower bond surface temperature of −25 °C and maximum porosity of 2.10% in bituminous concrete specimens, which does not meet the specification requirement of less than 2%. The bending resistance of asphalt concrete samples met the specification requirements at −10 °C, 10 °C, 30 °C, 50 °C, and 70 °C at lower bonding temperatures but was stable at lower bonding temperatures of 10 °C, 30 °C, 50 °C, and 70 °C. The results show that the critical temperature of the interlayer bond is −10 °C when the temperature of the lower the interlayer bond surface is −10 °C.(2)The bending stress, strain, and deflection of bituminous concrete core wall increase with the increase of the bond surface temperature, especially when the bond surface temperature is less than −10 °C. Through the specimen after fracture cross-section, it can be seen that, when the asphalt concrete lower bond surface is at 10 °C or below, the specimen has brittle damage, and the upper layer of the asphalt mixture with large-diameter aggregates can not be effectively embedded in the lower layer of the mixture, thus resulting in a more flat bond fracture cross-section.(3)According to the test analysis, the temperature of the combined surface of the lower layer of asphalt concrete is less than 10 °C, which does not guarantee the bonding mass between layers, and should be heated to 30 °C and not to 70 °C as required by the specification. If the temperature of the lower bond surface is above 10 °C, no heating measures are needed, which provides a theoretical basis for the construction of an asphalt concrete core wall under low temperature.

## Figures and Tables

**Figure 1 materials-16-04133-f001:**
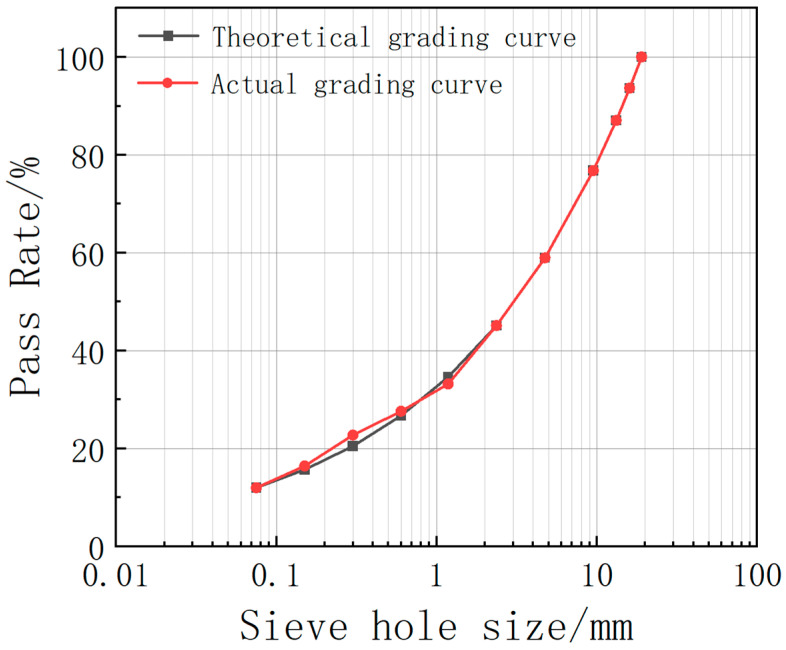
Aggregate grading curve.

**Figure 2 materials-16-04133-f002:**
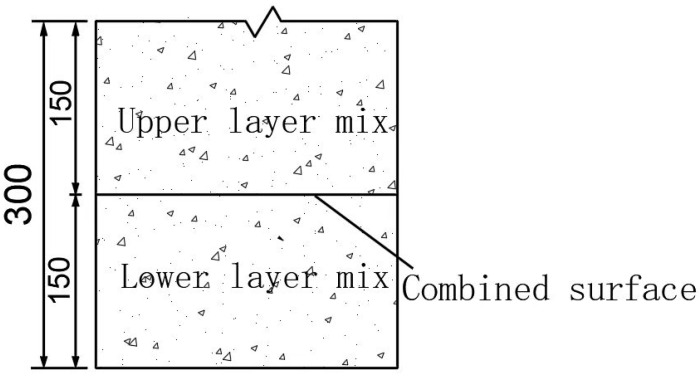
Schematic diagram of specimen forming.

**Figure 3 materials-16-04133-f003:**
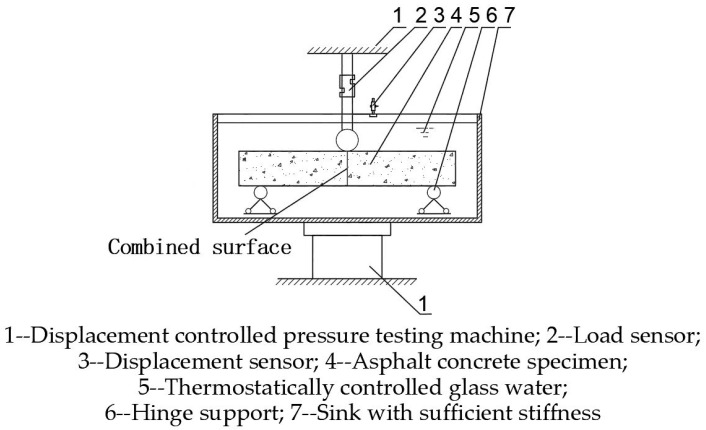
Schematic of trabecular bending test.

**Figure 4 materials-16-04133-f004:**
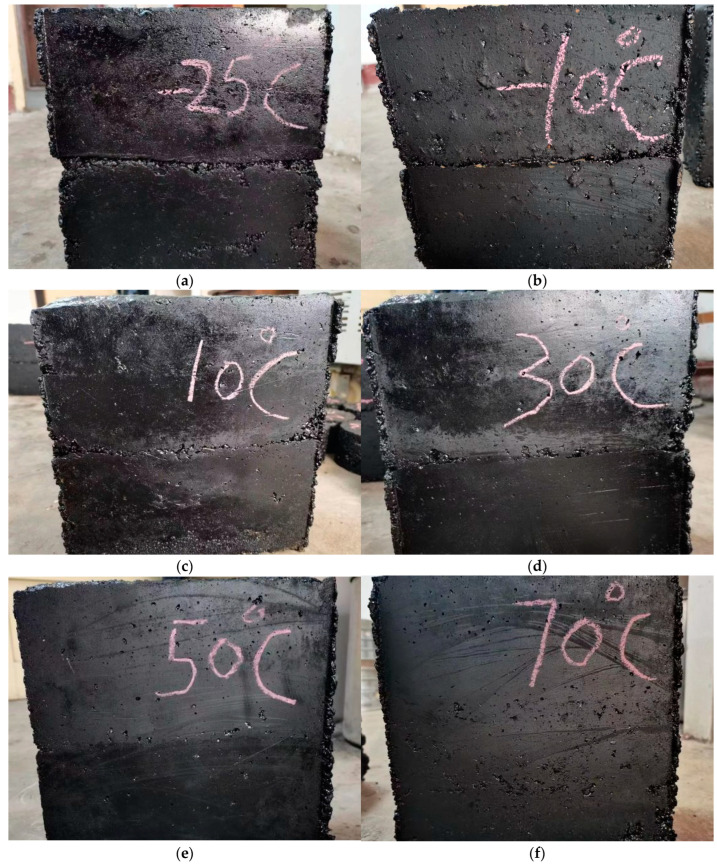
Asphalt concrete specimens at different bonding surface temperatures: (**a**) lower bond surface temperature of −25 °C; (**b**) lower bond surface temperature of −10 °C; (**c**) lower bond surface temperature of 10 °C; (**d**) lower bond surface temperature of 30 °C; (**e**) lower bond surface temperature of 50 °C; (**f**) lower bond surface temperature of 70 °C.

**Figure 5 materials-16-04133-f005:**
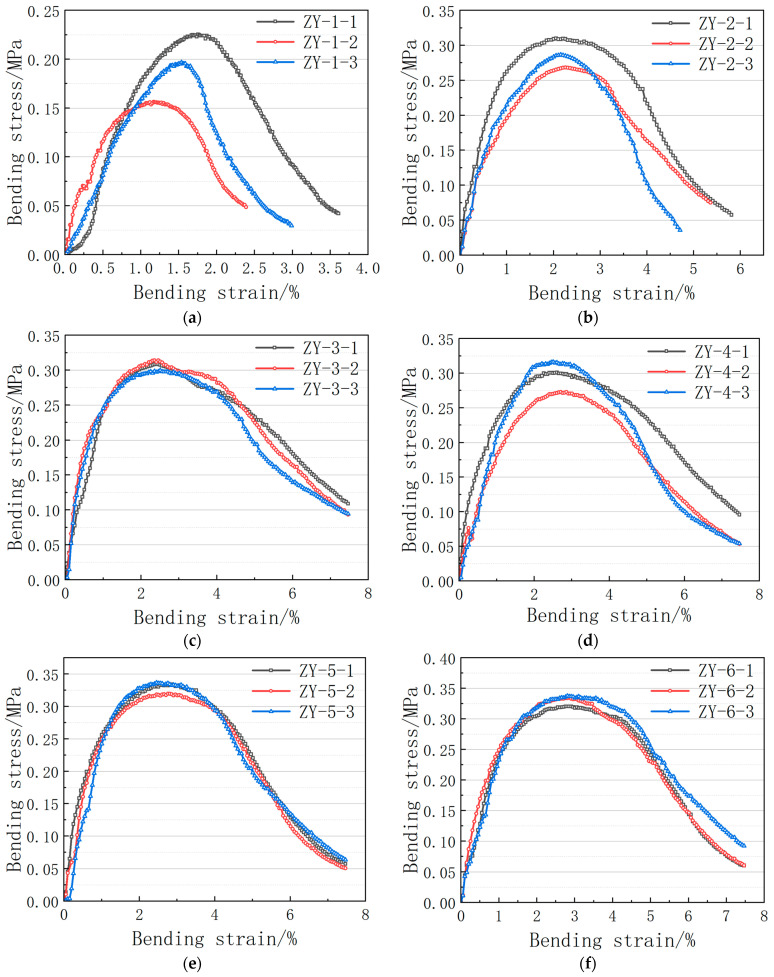
Bending stress–strain curves at different bond surface temperatures: (**a**) lower bond surface temperature of −25 °C; (**b**) lower bond surface temperature of −10 °C; (**c**) lower bond surface temperature of 10 °C; (**d**) lower bond surface temperature of 30 °C; (**e**) lower bond surface temperature of 50 °C; (**f**) lower bond surface temperature of 70 °C.

**Figure 6 materials-16-04133-f006:**
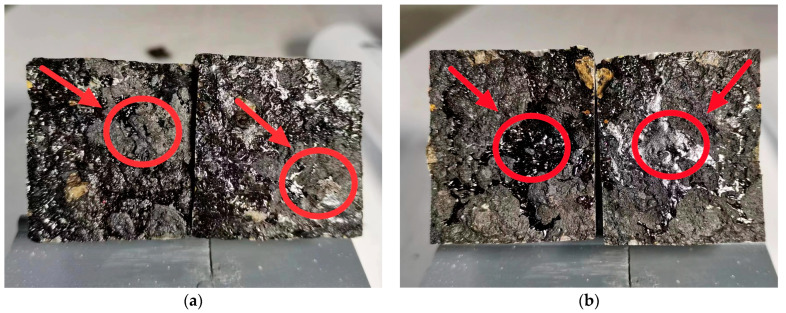
Fracture sections of small beam bending specimens at different bonding surface temperatures: (**a**) lower bond surface temperature of −25 °C; (**b**) lower bond surface temperature of −10 °C; (**c**) lower bond surface temperature of 10 °C; (**d**) lower bond surface temperature of 30 °C; (**e**) lower bond surface temperature of 50 °C; (**f**) lower bond surface temperature of 70 °C.

**Figure 7 materials-16-04133-f007:**
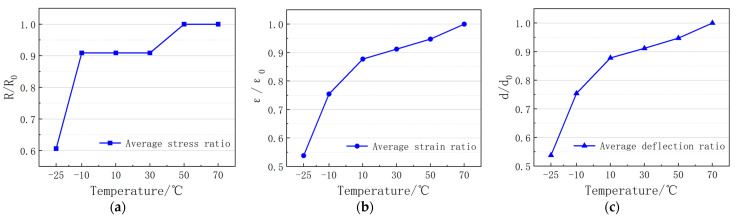
Curves of stress ratio, strain ratio, and deflection ratio with the temperature of the bonding layer: (**a**) stress ratio with the temperature of the bonding layer; (**b**) strain ratio with the temperature of the bonding layer; (**c**) deflection ratio with the temperature of the bonding layer.

**Table 1 materials-16-04133-t001:** Test results of coarse aggregate quality.

Technical Specifications	Specification Requirements	Test Results
Surface Density/(g/cm^3^)	≥2.6	2.710
Water absorption rate/%	≤2	0.48
Adhesion to asphalt/level	≥4	5
Crushing rate/%	≤30	12.55
Durability/%	≤12	1.3

**Table 2 materials-16-04133-t002:** Test results of fine aggregate quality.

Technical Specifications	Specification Requirements	Test Results
Surface Density/(g/cm^3^)	≥2.55	2.734
Water absorption rate/%	≤2	0.4
Water stability grade/level	≥6	9
Sodium sulfate 5 times cycle weight loss/%	≤15	3.4
Organic matter content	≤2	0

**Table 3 materials-16-04133-t003:** Filler quality test results.

Technical Specifications		Specification Requirements	Test Results
Surface Density/(g/cm^3^)		≥2.5	2.708
Water content/%		≤0.5	0.18
Hydrophilic coefficient		≤1.0	0.71
Packing grade sieving results/%	0.075 mm	>85	99.8
	0.15 mm	>90	100
	0.6 mm	>100	100

**Table 4 materials-16-04133-t004:** Quality test results of Karamay No.70 petroleum asphalt.

Projects		Quality Indicators	Test Results
Needle penetration(25 °C, 0.1 mm)		60~80	68.1
Crisp Point (15 °C)		≤−10	−20.8
Latency (15 °C, 5 cm/min)		≥150	>150
Latency (4 °C, 1 cm/min)		≥10	23.0
Softening point/°C		≥46	48.1
Solubility/%		≥99.5	99.9
Flash Point/%		≥260	310.0
Density 25 °C (g/cm^3^)		Actual test	0.986
Wax content/%		≤2	1.8
After the film oven	Quality change/%	±0.8	−0.10
	Residual needle penetration ratio (25 °C)/%	≥61	80.6
	Latency (10 °C)/cm	≥6	28.3

**Table 5 materials-16-04133-t005:** Trabecular bending test results at different joint surface temperatures.

Combined Surface Temperature/°C	Specimen Number	Density/ (g·cm^−3^)	Porosity/%	Deflection/mm	Maximum Bending Strength/MPa	Maximum Bending and Pulling Strain/%
−25	ZY-1-1	2.381	2.10	2.92	0.23	1.7535
	ZY-1-2	2.393	1.59	1.95	0.16	1.1690
	ZY-1-3	2.381	2.10	2.56	0.20	1.5364
	Average value	2.385	1.93	2.56	0.20	1.5364
−10	ZY-2-1	2.396	1.45	3.42	0.31	2.0541
	ZY-2-2	2.402	1.22	3.76	0.29	2.2545
	ZY-2-3	2.397	1.41	3.59	0.29	2.1543
	Average value	2.398	1.36	3.59	0.30	2.1543
10	ZY-3-1	2.403	1.19	4.01	0.31	2.4048
	ZY-3-2	2.411	0.85	4.09	0.31	2.4549
	ZY-3-3	2.401	1.28	4.18	0.30	2.5050
	Average value	2.405	1.19	4.18	0.30	2.5050
30	ZY-4-1	2.408	0.96	4.34	0.30	2.6052
	ZY-4-2	2.403	1.20	4.51	0.27	2.7054
	ZY-4-3	2.401	1.26	4.18	0.32	2.505
	Average value	2.404	1.20	4.34	0.30	2.6052
50	ZY-5-1	2.402	1.22	4.76	0.32	2.8557
	ZY-5-2	2.406	1.07	4.68	0.33	2.8056
	ZY-5-3	2.407	1.02	4.09	0.34	2.4549
	Average value	2.405	1.10	4.51	0.33	2.7054
70	ZY-6-1	2.387	1.83	4.84	0.32	2.9058
	ZY-6-2	2.388	1.80	4.76	0.33	2.8557
	ZY-6-3	2.398	1.38	4.68	0.34	2.8056
	Average value	2.391	1.80	4.76	0.33	2.8557

## Data Availability

Not applicable.

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
