# Peer review of "Effect of Interlayer Bonding Temperature on the Bending Properties of Asphalt Concrete Core Wall"

_materials, 2023, doi:10.3390/ma16114133_

Round 1
Reviewer 1 Report
It is an interesting article with industrial application, they focus on the influence of interlayer bonding temperature on the bending performance of asphalt concrete core wall. They conducted bending tests of small beams with different interlayer bonding temperatures and studied bending at 2 °C and focus on the analysis of the effect of lower bonding surface temperature change on their bending performance. The results they found are of great interest as the flexural stress, strain, and deflection of the asphalt concrete core wall increase with lower bond surface temperature, and the effect of temperature on asphalt concrete is particularly evident when the lower bonding surface temperature is below -10°C.
The article has a contribution but there are some observations.
1.- It is necessary to add in the introduction section a paragraph that describes how your work is sectioned. This must go to the end.
2.- What is the main question addressed by the research?
3.- Tables 1-4 must go in an annexes section.
4.- Figure 1 must have better quality and the size of its letters must be larger.
5.- Figure 2 must have better quality
6.- Figure 3 must have better quality, they are not appreciated with quality.
7.- Increase the size of the letters in figure 5. They cannot be observed.
8.- Increase the size of the letters in figure 7. They cannot be observed.
9.- Are the conclusions consistent with the evidence and arguments presented and do they address the main question posed?
10.- It is necessary to add more references.
After having reviewed this work, it is necessary to address the observations so that it can be accepted.
Reviewer 2 Report
A native English speaker must proofread the article.
The topic of the article is not innovative. Therefore, the contribution to the state of knowledge is low. The issue of bonding between layers of bituminous mixtures at low temperatures is already well-developed in the literature. Nevertheless, the results may be a helpful contribution to the case study presented because the bituminous mixtures are particularly dense.
The literature review in the introduction is poor concerning the fundamentals of the phenomena studied in the case study. The references indicated are generally relatively old, showing some lack of depth on phenomena studied in the introduction.
The article is well structured, but the development of the text lacks clarity. The test standards considered to characterize the base materials are not mentioned.
Sections 3.1 and 3.3 do not have adequate content for the intended purpose. Figure 4 is presented without any introductory text before its presentation.
Therefore, I suggest the authors improve the article in the indicated aspects before resubmitting it.
Reviewer 3 Report
This paper summarizes a study done on the bonding temperature on the bending properties of the asphalt material used in asphalt core walls. The text is difficult to understand and many instances. Major English revision is required; many sentences are too long. The objective of the study is not clear enough and the literature review does not support the need for the objective that is stated.
The use of bending tests to review the interlayer bounding is a good choice, but it is not clearly explained why this test, and the specific testing conditions were selected. Also, It would be helpful to have a general overview of the full testing program at the beginning of section 2.
The analysis is interesting, but incomplete. The energy before breakage would give interesting information. Also, the variability of the results could be used since it seems that there is more variability at lower temperature.
Here are specific comments:
- The literature review is complete enough, but it reads like a list. The text does not flow well.
- L101, what is Karamey No. 70 petroleum asphat? It is not a common name for me
- Table 1 to 4, indicates which standards are used for the specs
- 2.3 – “Material matching ratio”, this is mix design I guess
- L122, explain mineral grading index
- Fig 1. Text is too small
- 3.1, need more information. What is the precision of the readings?
- L135, … concrete were compacted 7 times each for one time, 15 times and 105 times in total… Please give more details. The specimen preparation is important. Was the same compaction procedure used for upper and lower mix?
- L139-140, what is the variability of the temperature?
- L144-145, so the specimens were compacted on top of one another, but then the interface is vertical during the tests? Why those specific dimensions?
- Fig 2, the legend should be in the figure and not below it
- Fig 3 is not called in the text and should be before fig 2. It would help to see how the test specimen are cut from the compacted slabs
- Fig 4 should appear after it’s mentioned in the text
- L177, we do not see if the bonding is good or not. We can just see that there is an interface that can be glued properly or not
- L188-190, sentence unclear
- Table 5, porosity of what? The interface? Measure how?
- Table 5, MPa and not Mpa
- Table 5, density of what? It must be the top specimen, but the density near the interface?
- L210, 211, 1,93% is smaller than 2%, so I do not understand
- L228, …is relatively flat,.. what do you mean?
- Figure 7, define the stress ratio
Reviewer 4 Report
The manuscript presents a study on the the effect of interlayer bond temperature on the flexural performance of asphalt concrete core walls. However, the authors need to address the following issues to improve the quality of the manuscripts for publication.
1. Lines 8-13: This sentence appears too complex and resulted in distorted understanding. The authors need to split the sentence into 2 or 3 sentence to aid better understanding of the reader.
2. Lines 11-12: “ …performance of asphalt concrete core wall;By making small beam bending specimens with different interlayer bonding temperatures, Bending test study at 2°C,And focus on analyzing the…” The authors were using capital letter to begin a word after a comma. This is incorrect and should be corrected.
3. The aim of the study was not stated in the Abstract section. The authors need to state the aim of the study before the results part of the Abstract.
4. The authors need to also add to the abstract a brief or summary of the methodology employed in the study to help the reader understand the results presented in the Abstract.
5. The authors need to work on the manuscript to correct grammatical errors. They could also make use of the free version of the Grammarly software to do this.
6. Line 20: “The lower bonding surface temperature should be heated to 30°C’’. Is this a recommmendation? If yes, what is the basis?
7. Line 27 and 33: The authors need to avoid the use of lumped-up references such as [1-4], [11-18] and [19-23]. They need to give a brief summary of the findings of the references by limiting the numbers of references for a comment/sentence to be one or two or at most three references.
8. Why was the majority of the manuscript contents highlighted in yellow. The yellow painting is not friendly to the eyes. The authors should be advised against highlighting the majority of the manuscript texts in yellow because it affected my review of the manuscript.
9. Lines 30-40: This sentence appears to be too long (a sentence taking up 10 lines is not ideal). The authors need to work on the sentence structure of their manuscript contents to reduce the complexity of the sentences.
10. Line 96: ‘analysis’ was written twice, the authors should delete one.
11. Table 1: No values was given for the heat resistance but the authors indicated qualified. On what basis is the heat resistance qualified?
12. Tables 1, 2 and 3: The authors need to reference the source (s) of the specific requirements provided in Tables 1, 2 and 3.
13. Table 4: The authors need to reference the source (s) of the quality indicator provided in Table 4.
14. Lines 224-232: This sentence needs to be restructured to aid the understanding of the reader as well as improve the quality of the manuscript.
Author Response
请参阅附件。

Round 2
Reviewer 2 Report
A native English speaker must proofread the article.
I think the changes performed are insufficient to achieve a suitable quality. The cover letter does not describe the changes made and the justification behind the changes.
The literature review remains relatively poor.
Author Response
请参阅附件。
